# BIT-WISE TRAINING OF NEURAL NETWORK WEIGHTS

## ABSTRACT

We propose an algorithm where the individual bits representing the weights of a neural network are learned. This method allows training weights with integer values on arbitrary bit-depths and naturally uncovers sparse networks, without additional constraints or regularization techniques. We show better results than the standard training technique with fully connected networks and similar performance as compared to standard training for residual networks. By training bits in a selective manner we found that the biggest contribution to achieving high accuracy is given by the first three most significant bits, while the rest provide an intrinsic regularization. As a consequence we show that more than 90% of a network can be used to store arbitrary codes without affecting its accuracy. These codes can be random noise, binary files or even the weights of previously trained networks.

## 1 INTRODUCTION

Many challenging areas of computer science have found very good solutions by using powerful techniques such as deep neural networks. Their applications range now from computer vision, speech recognition, natural language processing, game playing engines, natural sciences such as physics, chemistry, biology and even to automated driving. Their success is largely due to the increase in computing power of dedicated hardware which supports massive parallel matrix operations. This enabled researchers to build ever growing models with intricate architectures and millions or even billions of parameters, with impressive results.

However, despite their effectiveness, many aspects of deep neural networks are not well understood. One such aspect is why over-parameterized models are able to generalize well. One of the important avenues of research towards a better understanding of deep learning architectures is neural network sparsity. Frankle & Carbin (2019) showed a simple, yet very effective magnitude based pruning technique capable of training neural networks in very high sparsity regimes while retaining the performance of the dense counterparts. This sparked new interest in parameter pruning and a large body of work on the topic has since been published. The techniques for weight pruning can be broadly categorized as follows: pruning after training, before training and pruning during training. The work of Frankle & Carbin (2019) falls in the first category because the method relies on removing the weights which reach small magnitudes after they have been trained. In the second kind of approach, such as (Lee et al., 2019; Wang et al., 2020), neural networks are pruned before training in order to avoid expensive computations at training time. The end goal is to remove connections such that the resulting network is sparse and the weights are efficiently trainable after the pruning procedure. The third kind of approach is to use dynamical pruning strategies (Dai et al., 2019; Mostafa & Wang, 2019) which train and remove weights at the same time.

The main goal behind these pruning strategies is to find sparse neural networks which can be trained to large degrees of accuracy. However, it has been shown by Zhou et al. (2019) that there exist pruning masks which can be applied to an untrained network such that its performance is far better than chance. Furthermore, Ramanujan et al. (2019) developed an algorithm for finding good pruning masks for networks with fixed, random weights. Theoretical works (Malach et al., 2020; Orseau et al., 2020) even proved that within random neural networks there exist highly efficient subnetworks, which can be found just by pruning. Orseau et al. (2020) advance the hypothesis that the main task of gradient descent is to prune the networks while on the second place is the fine-tuning of the weights.

## 2 MOTIVATION

A key issue we want to emphasize is that, in all these works, the way in which the networks are pruned in practice is by forcing them, through some criteria, to set a fraction of the weights to zero. Since it has been shown that sparse networks perform as well as their dense counterpart, or sometimes even better, the natural question that arises is: *why doesn't gradient descent itself prune the weights during training?* Why hasn't pruning been spontaneously observed in practice? One possible explanation is that, at least for classification tasks, the usual cross–entropy loss without additional regularization techniques are not well suited for this. Other factors such as the stochasticity of the data batches, optimization algorithm, weights initialization etc. might also play a role.

However, we approach this question from a different perspective. We hypothesize that an important reason for weights not being set to zero is because this is a particular state where the bits representing a weight must all equal zero. This is highly unlikely since weights are usually represented on 32 bits. The probability of a single weight being set to exactly zero is $2^{-31}$, the sign bit not playing a role. Therefore the chances that a significant number of weights is set to zero decreases very rapidly. If weights would be represented on a lower bit depth, then the chance that the optimizer sets them to zero should increase.

In order to test the degree to which this hypothesis is true we experiment with neural networks for image classification where, instead of training the weights themselves, we train the individual bits representing the weights. This might allow gradient descent to reach stable states where all bits in a set of weights are zero and the loss function is around a local minimum. If our hypothesis is true then we expect a strong dependency between sparsity and bit-depth.

By encoding weights on arbitrary precision we also touch upon the topic of network quantization and show that particular cases of this training technique result in algorithms developed in previous works which we will describe in Section 8. Moreover, we show that weight quantization naturally leads to weight pruning and sparse networks without additional constraints such a regularizations, additional loss terms, architectural changes or other tricks usually involved in engineering low bit quantized networks.

## 3 BINARY DECOMPOSITION

We approximate weights on $k$ bits by using the sign and magnitude representation, due to its simplicity. A weight tensor of a layer $l$ can be decomposed as:

$$\theta_k^l = \left( \sum_{i=0}^{k-2} a_i^l \cdot 2^{i+\alpha_l} \right) \cdot (-1)^{a_{k-1}^l} \tag{1}$$

with $a^l \in \{0, 1\}$ representing the binary coefficients and $k$ the number of bits. The summation encodes the magnitude of the number while the second factor encodes the sign: this way we obtain numbers in a symmetric interval around zero. We add a negative constant $\alpha_l$ to the exponent in order to allow the representation of fractional numbers (see Table 1). Additionally, this term controls the magnitude of the numbers and, therefore, the width of the weights distribution within a layer. Choosing $\alpha_l < -k + 1$ the weights are guaranteed to be less than 1 in magnitude. In order to constrain $a$ to take binary values we use auxiliary floating point variables $x \in \mathbb{R}$ (virtual bits) passed through a unit step function: $a = H(x) = 0$ if $x \leq 0$, otherwise 1.

The weight initialization for the $k$ bit training technique is as follows: for a fully connected layer the weight matrix is expanded into a 3D tensor of shape $(k, n_{l-1}, n_l)$ with $k$ representing the number of bits and $n_{l-1}, n_l$ the number of nodes in the previous and current layer, respectively. Figure 1 illustrates a simple example of a $(3, 4, 3)$ bit-tensor. For convolutional layers, where a weight tensor is in higher dimension, the procedure is analogous to the fully connected case and the bit-tensor is now of shape $(k, s_x, s_y, n_{l-1}, n_l)$ with $s_x, s_y$ representing the kernel sizes in $x$ and $y$ direction. The value for each bit is chosen randomly with equal probability of being either 0 or 1 ($x \leq 0$ in the first case and $x > 0$ in the second). We ensure the weights sampled in this manner are not initialized at exactly zero, because this would mean pruning the network from the start and invalidate our hypothesis. Hence we obtain a uniform weight distribution without zeros. We adopt

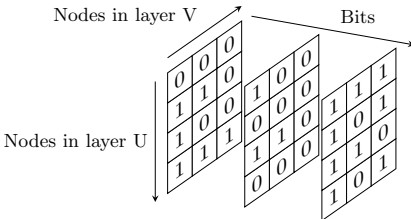

Figure 1: 3D bit-tensor connecting two dense layers, U and V, with 4 and 3 nodes, respectively.

the Kaiming He (He et al., 2015a) initialization technique for each layer's weights, which means the standard deviation is $\sqrt{2/n_{l-1}}$, where $n_{l-1}$ is the number of nodes in the previous layer. We have determined $\alpha_l$ algorithmically via a simple binary search such that this condition is fulfilled for the weight distribution of each layer. This term is a fixed parameter in each layer and depends only on the structure of the network. The virtual bits, $x$, are chosen from a normal distribution which also satisfies the Kaiming He condition on its variance. For the particular situation where $k = 2$ the weights have only two values and the standard deviation is exactly $2^\alpha$. Ramanujan et al. (2019) refer to this distribution as the Signed Kaiming Constant.

During training, the feed-forward step is performed as usual, with the weights being calculated according to Eq. (1). The backpropagation phase uses the straight through estimator (STE) (Hinton, 2012; Bengio, 2013) for the step function introduced in the weight's binary decomposition. The derivative of a hard threshold function such as the Heaviside step function is zero everywhere except at zero (more specifically it is the Dirac delta function). Since the values of the weights are passed through this step function are almost never exactly zero, the gradients during backpropagation will almost always be zero. This situation leads to a stagnant network which never updates its weights and never learns. To avoid this, during the backpropagation phase the gradient of the step function is replaced by the gradient of a different function which is non-zero on a domain larger than for the step function. Such functions are usually referred to as proxy functions and can take many forms. Yin et al. (2019); Shekhovtsov & Yanush (2020) provide in-depth discussions on the properties of STEs. Throughout this work we adopt the method first proposed by Hinton (2012) which treats the gradient of a hard threshold function as if it were the identity function. This method has been shown to work very well in practice (Liu et al., 2018; Bulat et al., 2019; 2021; Bethge et al., 2019; Alizadeh et al., 2019)

Notice that in Eq.(1) the additive constant $\alpha_l$ can be factored out of the sum. The resulting weights are in the form $\theta_k^l = 2^{\alpha_l} \cdot \Theta_k^l$, where $\Theta_k^l$ contains only integer numbers on $k$ bits. The ReLU activation function has the property that $\sigma(\alpha \cdot x) = \alpha \cdot \sigma(x)$ for any $\alpha > 0$. It can be shown that for a ReLU network of depth $L$, scaling the weights of each layer by a factor $\alpha_l$, with $l \in [0, 1, \ldots L-1]$ representing the layer index, is equivalent to scaling just a single layer with $\alpha = \prod_{l=0}^{L-1} \alpha_l$, including the input layer. This means that we can gather all factors $\alpha_l$ into a single $\alpha$, scale the input images with that factor and train the network with just integer numbers represented on $k$ bits. At inference time, for classification tasks $\alpha$ is not relevant because the output nodes are all scaled by the same coefficient and $\arg\max(\alpha \cdot \mathbf{x}) = \arg\max(\mathbf{x})$ for any $\alpha > 0$.

## 4 EXPERIMENTS

We have performed an extensive set of experiments where networks were trained on bit-depths ranging from 2 to 32. Figure 2 summarises the performance of LeNet and ResNet-18 (LeCun et al., 1998; He et al., 2015b) trained on MNIST and CIFAR10 (LeCun & Cortes, 2010; Krizhevsky, 2009). Each experiment was repeated 15 times. Each data point represents the best accuracy/sparsity obtained from all runs and are displayed as violin plots. They show via the kernel density estimation the minimum, mean, maximum and the spread of the repeated runs. The right-most violin shows the performance of the standard 32-bit training technique, the horizontal black line its mean and the shaded area the minimum and maximum accuracy.

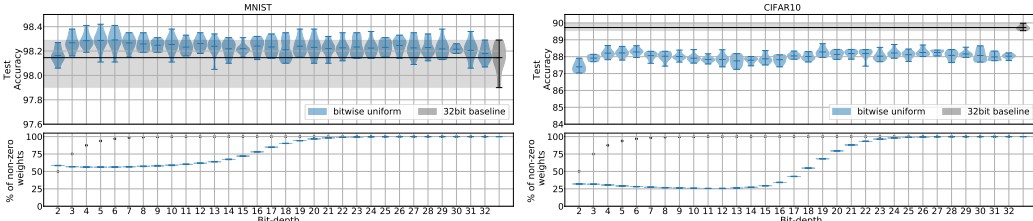

Figure 2: Classification accuracy and sparsity as a function of the weights bit-depth with LeNet (left) and ResNet (right). The right-most data point indicates the performance of the standard 32-bit training technique. Lower panels indicate the amount of non-zero weights remaining after training. Black dots show the probability that a certain percentage of weights is set to zero by random chance.

The networks were trained with the following setup. For LeNet the learning rate starts at $9 \cdot 10^{-4}$ and is divided by 10 at epoch 40 and 80. We have also experimented with a single, fixed learning rate but in that case the standard training technique on 32bits reached a maximum accuracy of only 97.7%, while bit-wise weight training did not suffer any noticeable penalty. For ResNet the learning rate starts at $6 \cdot 10^{-4}$ and is divided by 10 at epoch 150 and 170. In both cases we used the Adam optimizer (Kingma & Ba, 2017).

For LeNet (left panels in Figure 2) this training technique consistently achieves higher mean accuracies than the baseline while at the same time pruning the network significantly. Moreover, as the bit depth decreases there seems to be a slight increase in the mean classification accuracy. This indicates that the additional bits available for the weights impede the ability of the gradient descent to converge to better solutions. The right panels in Figure 2 show the results of ResNet-18 trained on CIFAR10. Here we observe a degradation in terms of classification accuracy compared to the standard training technique of about 1.7 percentage points (we will show in Section 5 how to mitigate this issue). The network sparsity is higher than in the case of LeNet, somewhere in the range of 25-35%, for bit depths 2 to 16. Note that the sparsity plots are also represented as violins, but their height is smaller relative to the scale of the entire curve due to the very small variations in the sparsity achieved at the end of training.

For both LeNet and ResNet there is a strong dependency between the bit depth and the amount of zero weights found by the network. This is in line with our hypothesis that gradient descent does not naturally uncover sparse networks when training weights represented on 32bits. This also explains why currently used pruning techniques require external mechanisms which force the network to set weights to zero while training. In essence, they bypass the weight's whole bit structure, effectively setting all bits to zero at once.

The black dots in Figure 2 indicate the percentage of weights set to zero by random chance. We observe that for high bit-depths ($k > 24$) the chance that gradient descent sets a certain amount of weights to zero is almost the same as random chance. However, for lower bit-depths gradient descent is much more likely to set weights to zero due to the much smaller search space of the weight's bit structure.

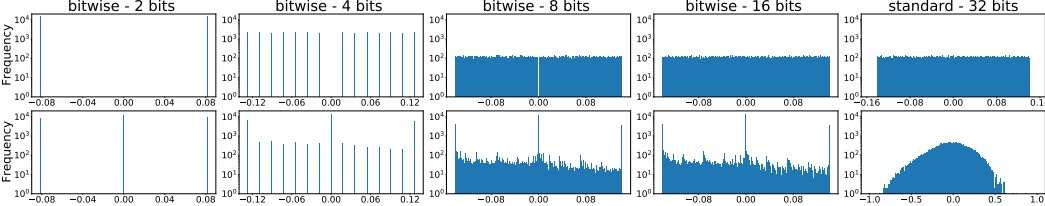

Figure 3: Weight distribution of the second LeNet layer before and after training.

Figure 3 shows the histogram of (float) weight distribution of the second hidden layer in LeNet before and after training. Bit-wise weight learning moves a significant amount of weights either to exactly zero or to the maximum value representable on $k$ bits. The frequency of intermediary

values is significantly reduced, in some cases by one order of magnitude. Although this technique has no special regularization nor an external weight pruning mechanism, it naturally uncovers sparse networks. This comes in stark contrast with the standard training technique, right most panels. Here, the distribution of the weights after training is much more spread out than the initial one and has a large peak towards zero, but the weights are never exactly zero.

# 5 SELECTIVE BIT TRAINING

In Section 4 we have presented experiments where all weight bits are simultaneously trained together. Our algorithm, however, also allows us to train specific bits only, while keeping others fixed as they were originally initialized. We can encode as a string mask of 0's and 1's which bit is trainable and which not, e.g. for a 4-bit mask **0001** we initialize all bits randomly but we train only the least significant bit, while for **1000** we train only the sign bit and leave the rest unchanged. See Table 1 for an example of a weight represented as a 16-bit number.

Table 1: Sign and magnitude representation of a number. Left-most bit gives the sign while right-most is the least significant bit. Using Eq. (1) and choosing $\alpha = 0$ results in a 16-bit signed integer while for $\alpha = -15$ results in a floating point number with a magnitude less than 1.

| Bit index | 15 | 14 | 13 | 12 | 11 | 10 | 9 | 8 | 7 | 6 | 5 | 4 | 3 | 2 | 1 | 0 |
|---|---|---|---|---|---|---|---|---|---|---|---|---|---|---|---|---|
| **Bit value** | **1** | **0** | **0** | **1** | 0 | 0 | **1** | 0 | **1** | 0 | 0 | 0 | **1** | **0** | **0** | **0** |
| **Trainable** | 1 | 1 | 1 | 1 | 0 | 0 | 0 | 0 | 0 | 0 | 0 | 0 | 1 | 1 | 1 | 1 |
| **Significance** | sign | \multicolumn magnitude | | | | | | | | | | | | | | |
| **W** ($\alpha = 0$) | -4744 | | | | | | | | | | | | | | | |
| **W** ($\alpha = -15$) | -0.144775390625 | | | | | | | | | | | | | | | |

Figure 4 show the results achieved by LeNet with all possible selective training patterns for 2, 4 and 8 bits. Training with weights encoded on 2 bits (top-left panel) results in 3 possible scenarios: '01' trains the magnitude, '10' trains the sign and '11' trains the sign as well as the magnitude of the weights. Training with weights encoded on 4 bits, pattern '1000' corresponds to training just the sign and keeping the magnitudes random, '0111' corresponds to training the magnitudes and keeping the sign fixed and '1111' corresponds to training all bits. Similarly for 8 bits (bottom panel). The baseline accuracy is shown as the right-most data-point in each graph.

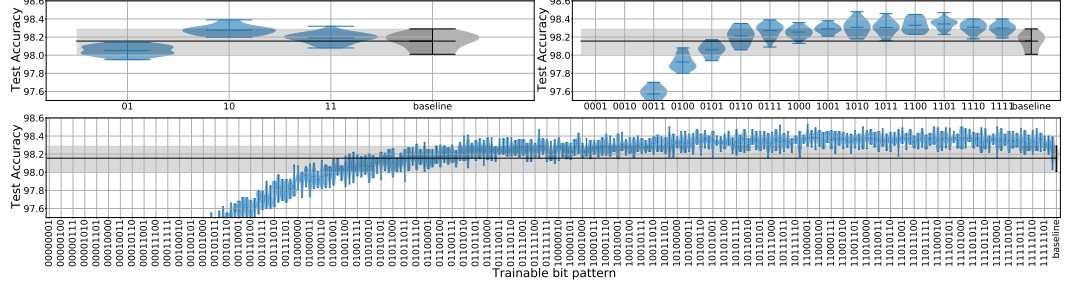

Figure 4: LeNet trained with all bit patterns for 2 (left), 4 (right) and 8 bits (bottom panel).

Figure 5 shows the same experiments for ResNet. An interesting phenomenon appears when training bits selectively. Several strong discontinuities in the accuracy curve are visible when training weights encoded on 4 and 8 bits. They appear at very specific bit patterns which we will address next. First, we highlight the extreme situations of (*a*) training just the sign bit and (*b*) only the magnitude bits. In Figures 4 and 5 these refer to the central data points with trainable bit patterns '10', '1000', '10000000' for sign training and '01', '0111', '01111111' for magnitude training.

When training just the sign bit, LeNet outperforms the baseline network, as shown in Figure 5. Our weight initialization procedure avoids initializing magnitudes to zero. For the particular case when quantizing weights on $k = 2$ bits it means that the magnitude bit is always 1. In this situation training only the sign bit is therefore equivalent to training a binary network with $\Theta \in \{-1, 1\}$. For

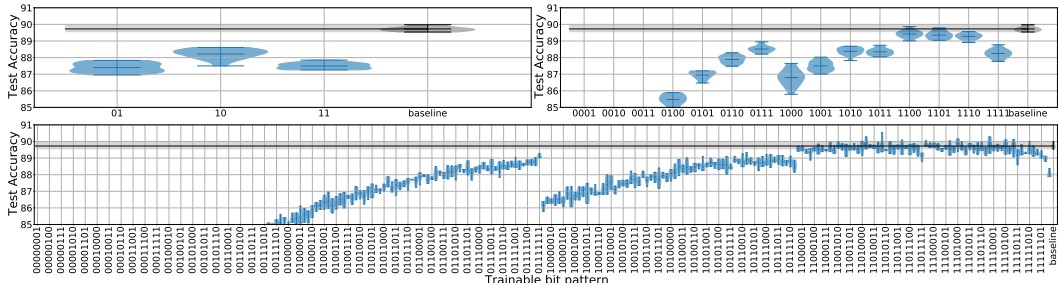

Figure 5: ResNet trained with all bit patterns for 2 (left), 4 (right) and 8 bits (bottom panel).

ResNet, Figure 5, training the weight's sign leads to a performance drop of 2–4 percentage points, depending on the quantization size. It shows that this particular network can be trained reasonably well only by changing the sign of the weights and never updating their magnitudes.

Training only the magnitude bits results in a very small performance penalty for LeNet as compared to the baseline, and about 1–3 percentage points for ResNet. Training all bits simultaneously leads to the average performance between the two extreme cases. This phenomenon is valid for both ResNet and LeNet, although less visible for the latter. We have performed experiments for bit depths ranging from 2 to 32, where we train only the sign and only the magnitude bits in ResNet. Figure 6 summarizes the test accuracy and sparsity obtained in these two cases. Notice there is little to no correlation between accuracy and bit-depth above 8, whereas sparsity is strongly influenced by it, particularly above 14. For bit-depths lower than 5, magnitude only training rather decreases in performance, while sign only training increases. For the extreme $k = 2$ bits quantization, their accuracy ordering is inverted and in this case training both the sign bit and the magnitude bit results in a ternary network with $\Theta \in \{-1, 0, 1\}$.

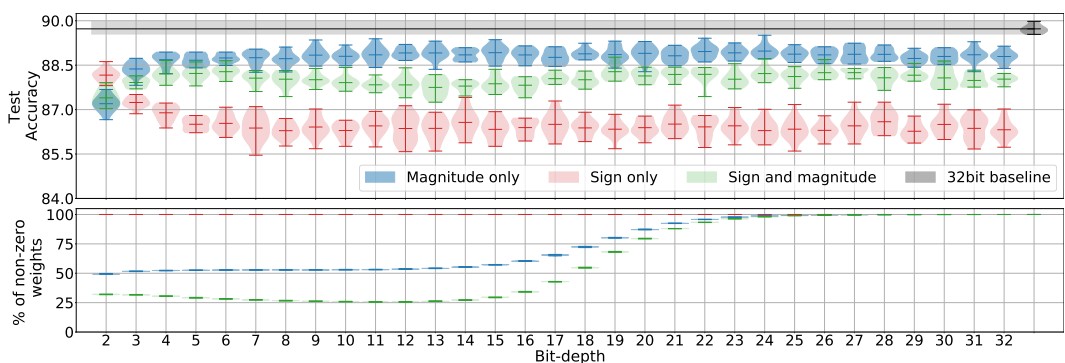

Figure 6: ResNet test accuracy and sparsity after bit-wise training: only the magnitude bits (blue), only the sign bit (red), all bits (green). Baseline indicated by right-most data point.

The second important observation refers to the cases where the sign and the next one or two bits are trained, while the following remain randomly initialized. These situations correspond to the trainable bit patterns '1100', '1110', '11000000' and '11100000' in Figures 4 and 5. In all these cases the bit-wise training technique reaches an accuracy above the baseline (LeNet) or similarly to it (ResNet). This behaviour indicates that a fraction of the untrainable (and less significant) magnitude bits act as a regularizer, increasing the accuracy of the network as compared to the case when they are also trained. We investigated how many trainable bits would be sufficient to reach the accuracy of the baseline. To this end we perform bit-wise training on ResNet with 32, 16, 8, 6, 4 and 2 bits encoding for the weights and gradually decrease the number of trainable bits. More specifically, we expand Eq. (1) in the following way:

$$\theta_k^l = 2^{\alpha_l} \left( \sum_{i=0}^{p-1} \underbrace{a_i^l}_{\text{untrainable}} 2^i \ + \ \sum_{j=p}^{k-2} \underbrace{a_j^l}_{\text{trainable}} 2^j \right) \cdot \underbrace{(-1)^{a_{k-1}^l}}_{\text{trainable}} \tag{2}$$

where $k$ represents the weight's bit-depth and $p$ the number of untrainable bits. For $p = 0$ all bits are trainable and for $p = k - 2$ only the sign is trainable.

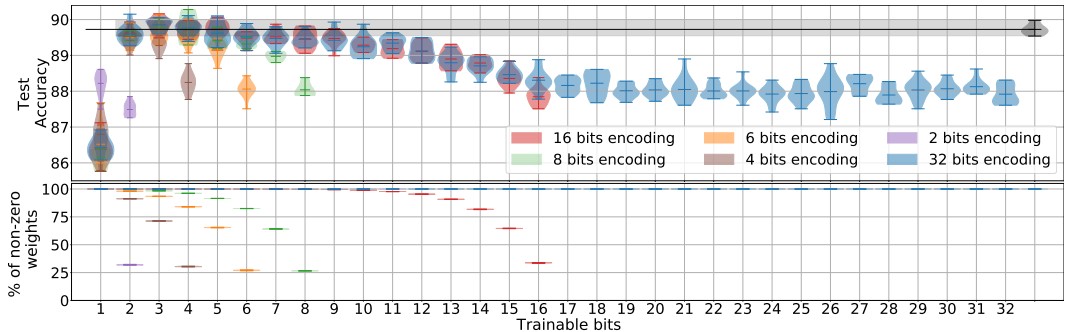

Figure 7: ResNet accuracy as a function of number of trainable bits, counting from the sign bit. Colors represent the bit-depth of the weights. The baseline, 32bit floating point weights, is shown as the right-most gray data point.

We summarize the results of these experiments in Figure 7. The blue data points represent the test accuracy of ResNet as a function of the number of trainable bits, with weights encoded as 32bit integers. Training more than 17 bits results in a test accuracy of about 88%. As the number of trainable bits decreases the accuracy improves and reaches the level of the baseline when training only the first 3 bits. A similar behaviour is seen when encoding weights on lower bit-depths. The best performance is obtained when weights are encoded on more than 6 bits and we train the sign and the next two most significant magnitude bits. The rest of the available bits do not contribute to the network's performance, rather they hinder the capacity of the network to converge well.

## 6   POST-TRAINING BIT ANALYSIS

Training bits selectively uncovers the fact that only a few of the most significant bits contribute to achieving a high accuracy, while the others provide regularization. In contrast, standard training does not reveal which weights or bits contribute most to the network's performance. In order to understand this we conduct experiments where we convert the weights learned in the standard way into weights expressed according to Eq. (1). More precisely, we start by training a standard network, and after training, for each layer we divide all weights by the magnitude of the smallest non-zero weight within that layer and round to the nearest integer. Therefore we obtain integer weights which we can then decompose into binary form and gain access to each bit. To be as close as possible to the original weights we encode the integer weights on 32 bits, even though in most situations weights do not require that many. Thus we convert a network trained in the standard way, weights as 32bit floating point values, into a network with integer weights on 32 bits.

Next, we start changing the first $p$ less significant magnitude bits and leave the next $32 - p$ bits unchanged, similar to Eq. (2). In this way we can investigate the impact of each bit on the final accuracy. Note that different layers require a different number of bits to represent the weights and generally, but not necessarily, depends on the number of weights within that layer. If we start changing more bits than a layer requires, the pre-trained structure is destroyed and the network looses its properties. In order to avoid this, we compute the maximum number of bits required for the weights in each layer, $m_l$. We impose that the maximum number of changed bits for each layer is $p_l^{max} = m_l - 3$.

Figure 8 shows the accuracy and sparsity of a standard, pre-trained LeNet and a 6 layer VGG-like network, Conv6, (same as Frankle & Carbin (2019); Zhou et al. (2019); Ramanujan et al. (2019))

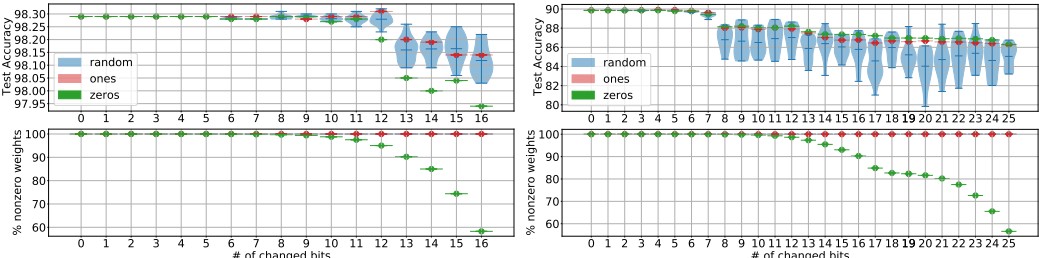

Figure 8: Test accuracy for LeNet (left) and Conv6 (right) as a function of the number of changed bits after training. Red points correspond to setting bits to 1; green correspond to 0; blue violins correspond to setting bits randomly to either 0 or 1. Bottom panels indicate sparsity.

as a function of the number of changed bits. We have experimented with three scenarios: all bits are changed randomly, all bits are set to 0, all bits are set to 1. The first data point in each graph, $p = 0$, represents the performance of the unmodified network with 32 bit floating point weights, as no bits are changed. The following entries indicate the performance of the network as we gradually increase the amount of changed bits. LeNet extends up to 16 bits (the maximum allowed for the first layer in this particular network) and Conv6 extends up to 25 (the maximum allowed for the first dense layer within this network). Setting all $p$ bits to zero (or one) leads to a single possible set of weights. Setting $p$ bits randomly leads to more possible outcomes. This difference is illustrated in Figure 8 by the way the data-points are represented: a single dot when setting bits to zero/one and a violin when setting bits randomly.

One can observe that also weights trained in a standard 32 bit floating point format do not make full use of high precision bits. The first 6 bits do not play a significant role for the final accuracy, as they can be modified post-training to any value. These results are in line with our initial hypothesis that gradient descent does not prune networks due to the large amount of bits available for the weights. Additionally, we found that the most important contribution to the performance of a network is the sign bit, followed by the next two most significant magnitude bits. This suggests that gradient descent might find a local optimum based only on these three bits while the rest are used to perform fine-tuning. However, this appears to be less successful, since a large fraction of the bits might be set to zero, one or left randomly initialized, perhaps due to the stochasticity of the training algorithm (batch training) or the noise present in the data itself.

## 7 MESSAGE ENCODING IN WEIGHTS

We have shown so far that 29 out of the 32 bits available for the weights of ResNet have an overall regularization behaviour and can remain randomly initialized and never trained. This leads to the idea that they could be used to encode arbitrary messages while the trainable bits are sufficient to train the network to high degrees of accuracy. To test this hypothesis we have performed several experiments in which we embedded various types of messages in the first 29 untrainable bits of a neural network's weights and train only the next 3. The results are summarized in Figure 9. Each experiment was repeated 10 times.

The first data point shows the baseline accuracy of ResNet trained with the standard method (32bit floating point representation of weights). For the second experiment we assigned random values to the untrainable bits of each layer. In the third experiment we embedded random passages of Shakespeare's Hamlet. In the fourth experiment we trained until convergence 29 ResNets with bit-depth 1 and embedded each of them into a new, 32bit ResNet, training in a bit-wise fashion the sign and the next two most significant magnitude bits. The test accuracy obtained by the 1bit ResNet is shown as the last violin. We observe that embedding either random noise, structured data or a set of previously learned weights does not impact the accuracy with respect to the baseline ResNet in any significant way.

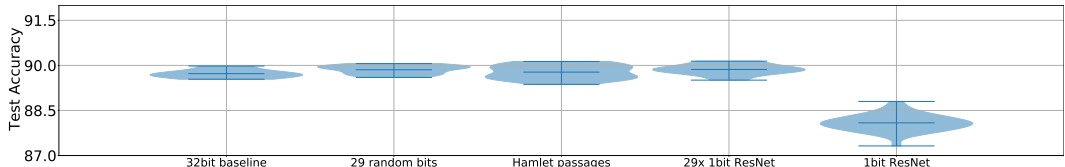

Figure 9: Test accuracy with various messages embedded in the first 29 untrainable bits of ResNet.

## 8 CONNECTION WITH OTHER WORKS

Our weight initialization procedure described in Section 3 ensures that weights are never set to zero before training. For the particular case where $k = 2$ bits this means that the magnitude bit is always 1 while the sign bit can be either 1 or 0. Training only the sign bit is therefore equivalent to training a binary network. This is similar to BinaryConnect, BinaryNet (Courbariaux et al., 2015; 2016) and XNOR-Net (Rastegari et al., 2016) where weights are constrained to $-1$ and 1. When training with bit pattern '01' (magnitude only) or '11' (sign and magnitude) results in a ternary network (Li & Liu, 2016; Zhu et al., 2017) because the magnitude is now also allowed to change, leading to some weights being set to zero.

Training only the magnitude bit the behaviour of our algorithm is effectively very similar in nature and performance to the *edge-popup* algorithm developed by Ramanujan et al. (2019) which finds pruning masks for networks with weights randomly sampled from the Signed Kaiming Constant distribution. Encoding weight on arbitrary bit depths and training just the sign bit we obtain the *sign-flipping* algorithm first shown by Ivan & Florian (2020).

Wang et al. (2021) found in a recent study that it is possible to embed 36.9MB of malware into the dense layers of a pretrained 178MB Alex-Net model with a $1\%$ accuracy degradation and without being detected by antivirus programs. Our method can store arbitrary code in any layer of a network (dense as well as convolutional) and could drastically increase the viral amount without damaging the network's performance, at the same time raising no suspicion on the presence of the malware.

## 9 SUMMARY

Motivated by the question of why gradient descent does not naturally prune neural connections during training, we developed a method to directly train the bits representing the weights. From this perspective we show that an important factor is the over-parametrization in terms of number of bits available for weight encoding. This also sheds some light into why networks with large amounts of weights are able to generalize well.

Our algorithm enables weight quantization on arbitrary bit-depths and can be used as a tool for bit level analysis of weight training procedures. We show that gradient descent effectively uses only a small fraction of the most significant bits, while the less significant ones provide an intrinsic regularization and their exact values are not essential for reaching a high classification accuracy. A consequence of this property is that, by using 32 bits for the weight representation, more than 90% of a ResNet can be used to store a large variety of messages, ranging from random noise to structured data, without affecting its performance.

## 10 REPRODUCIBILITY

The code used for the experiments carried out in this work will be made public at: https://github.com/iclr2022-2798/bit-wise-training

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
