# OpenReview forum: "Bit-wise Training of Neural Network Weights"
_ICLR.cc/2022/Conference — ICLR 2022 Submitted_

### Official Review · Reviewer_xT9U · 2021-10-24

**Correctness:** 3
**Technical Novelty And Significance:** 2
**Empirical Novelty And Significance:** 3
**Recommendation:** 5
**Confidence:** 4

**Main Review:**

Here are the strengths and weaknesses of the paper.

Strengths:
1. The paper is interesting and, in some sense, novel in that it analyze the network weights/bits and regularization in an interesting perspective. By decomposing a weight into separate bits, the function of each bit can be more easily observed and analyzed.
2. The authors conducted extensive experiments to demonstrate different phenomena from the bit-wise training idea.

Weaknesses:
1. In the second paragraph, Page 5, I guess it should be "as shown in Figure 4" instead of "as shown in Figure 5".
2. In second sentence in Sec 5.1 is quoted here: "In contrast, standard training does not reveal which weights or bits contribute most to the network’s performance". I didn't find the following paragraphs support this claim. In contrary, the paragraphs show the most significant bits contribute more to the performance than the others. In network quantization, e.g., 32-bit to 8-bit, one can also find that the least significant bits can be dropped off without hurting performance much.
3. In second paragraph of Page 4, "..., in general, neural networks can be trained well only by changing the sign of the weights and never updating their magnitudes". I don't think 4-percentage drop is small considering Cifar10 is a small dataset and ResNet-18 is relatively a large model. When using a smaller network, the performance drop might be big.
4. Throughout the paper (e.g., Sec 5.1 and Sec 6), one main claim of the paper is "a few of the most significant bits contribute to achieving a high accuracy, while the others provide regularization". This is true but not a significant observation from the perspective of network quantization. When network is big and contains redundancy, the network can be quantized to lower-bit one (e.g., 3 bit) with comparable performance (e.g., [1]). Compared with the full-precision model, the quantized model is well regularized.

Questions:
1. As the paper hypothesizes, the 32-bit model does not have much zero weights because the probability of an exactly zero-valued weight is very small (1e-31). If this is true, one should expect the trend between 2 and 14 bit-width in Figure 2 is exponential instead of flat. I didn't find an explanation in the relevant section.

[1] Zhang, Dongqing, et al. "Lq-nets: Learned quantization for highly accurate and compact deep neural networks." Proceedings of the European conference on computer vision (ECCV). 2018.

**Summary Of The Paper:**

This paper proposes a neural network training technique such that individual weight bits can be optimized separately. In detail, each weight is represented as a weighted sum of its bits weighted by powers of 2. In training, updating each bit $b$ is achieved by updating a floating-point number $x$ ($b=1, x > 0; b=0, x \le 0$).

By conducing extensive experiments, the authors find:
1) Network with shorter bit-width show more weight sparsity than that with longer bit-width. (Sec 4.)
2) With selective bit training, only a few most significant bits contribute to the final high model accuracy. The other less important bits serve as regularization. (Sec 5.)
3) The less significant bits can be used to encode other information. (Sec 6.)

**Summary Of The Review:**

The perspective of the paper is interesting, but some claims might need correction. Moreover, the observation is straightforward, especially from the view of network quantization.

---

> ### Author Response · Authors · 2021-11-19
> **Response to reviewer xT9U**
>
> "In the second paragraph, Page 5, I guess it should be "as shown in Figure 4" instead of "as shown in Figure 5".
> > We have corrected that in the text.
>
> "In second sentence in Sec 5.1 is quoted here: "In contrast, standard training does not reveal which weights or bits contribute most to the network’s performance". I didn't find the following paragraphs support this claim. In contrary, the paragraphs show the most significant bits contribute more to the performance than the others. In network quantization, e.g., 32-bit to 8-bit, one can also find that the least significant bits can be dropped off without hurting performance much."
>
> > Standard network quantization methods directly train weights on low bit-depths, e.g. 8 bits. This is not equivalent with saying that the network ignores the contribution of 24 bits out of 32, simply because the 24 bits are not there. In this setup we cannot conclude that the extra 24 bits did not contribute to the performance. When training directly on 8 bits it is not possible to trace back which of the "missing 24 bits" did and which did not contribute to the final performance.
> In contrast, the procedure described in Sec 5.1 quantitatively shows how many bits can be removed (or randomly changed) from 32bit floating point weights without degrading the performance, hence indicating which bits contribute more to the network's performance.
>
> "In second paragraph of Page 4, "..., in general, neural networks can be trained well only by changing the sign of the weights and never updating their magnitudes". I don't think 4-percentage drop is small considering Cifar10 is a small dataset and ResNet-18 is relatively a large model. When using a smaller network, the performance drop might be big."
> > "Training well" is, unfortunately, rather subjective and up for debate whether a 4% drop is small or not. It is, of course, significant and we did not state otherwise. But considering that we train only 1 out of 32 bits (~3% of the weight's bit structure) and reach 96% of the baseline performance we believe this is not an insignificant performance.
> Comparing the baseline performance of ResNet-18 and Conv6 (Figure 6 and 8) we see that both reach an accuracy of ~90%. However, Conv6 is a network with 1.7M parameters, while ResNet-18 is a network with a similar size to LeNet-300, about 270K parameters, 6 times smaller than Conv6. Thus, we already have a comparison between a large model and a significantly smaller one.
>
> As the paper hypothesizes, the 32-bit model does not have much zero weights because the probability of an exactly zero-valued weight is very small (1e-31). If this is true, one should expect the trend between 2 and 14 bit-width in Figure 2 is exponential instead of flat. I didn't find an explanation in the relevant section.
> > We have added a curve comparing the amount of weights set to zero by random chance with the amount of weights set to zero by gradient descent, all as a function of the bit-depth.

---

### Official Review · Reviewer_gsV7 · 2021-10-24

**Correctness:** 3
**Technical Novelty And Significance:** 3
**Empirical Novelty And Significance:** 3
**Recommendation:** 6
**Confidence:** 3

**Main Review:**

Strengths:
- This paper experiments with an interesting an intuitive idea.
- There are many interesting applications, e.g., more efficient networks and embedding hidden messages in network weights.
- Most figures include error bars.
- Figure 3 confirms a main hypothesis (fewer bits encourages sparser networks).

Weaknesses:
1) The paper could benefit from medium scale experiments, e.g., ImageNet. The method matches standard training on ImageNet but faces accuracy degradation on CIFAR. A concern is that this accuracy degradation would be even more substantial for problems such as ImageNet.
2) The paper would benefit substantially from a related work section. Since the paper is not 9 pages, there is definitely room for this. I am not an expert on quantization (perhaps another reviewer is) but I know that it is a very active research area. How does this papers method compare to standard methods in quantization? If the author's hypothesis is correct, networks trained with various quantization techniques should be sparse and it would be very interesting to verify this.
3) There is no discussion of how much extra compute / FLOPs is incurred by this method during training, which may be a drawback.

**Summary Of The Paper:**

This paper proposes to directly train the bit values of each parameter in a neural network, instead of directly optimizing the floating point value of each weight. By varying the number of bits that are allowed to be optimized, the authors show that with less bits the network will become automatically sparser. This method has many interesting applications, including fixing some bits to be a message and only training the rest.

**Summary Of The Review:**

The paper is very interesting but would benefit substantially from 1) medium scale experiments (e.g., ResNet on ImageNet) and 2) a more thorough discussion and comparison with related work.

---

> ### Author Response · Authors · 2021-11-19
> **Response to reviewer gsV7**
>
> "The paper could benefit from medium scale experiments, e.g., ImageNet. The method matches standard training on ImageNet but faces accuracy degradation on CIFAR. A concern is that this accuracy degradation would be even more substantial for problems such as ImageNet."
> > As shown in Figure 7 our method does not suffer from degradation for cases with bit depth larger than 3 and the first 3 most significant bits are trained. Moreover, our algorithm does not use regularizations, additional loss terms, architectural changes or other tricks usually involved in engineering low bit quantized networks. Due to the time (and resource) constraints for the revision of this work it is not realistic to for a sensible progress on experiments with ImageNet and larger networks.
>
> "The paper would benefit substantially from a related work section."
> > We added a dedicated section on related works.
>
> "If the author's hypothesis is correct, networks trained with various quantization techniques should be sparse and it would be very interesting to verify this."
> > Most papers on the topic of quantization do not correlate it with pruning. Our goal is not to perform  a comparative study of quantization vs. pruning literature. Rather it is to indicate a strong correlation between quantization and pruning from the perspective of binary representation of numbers used for weight encoding. Quantization vs. pruning is, indeed, a topic for future work.
>
> "There is no discussion of how much extra compute / FLOPs is incurred by this method during training, which may be a drawback."
> > There is surely a drawback to this method, as the number of parameters grows directly proportional to the number of bits used to encode the weights. On the GTX 1080Ti GPU that we used to do the experiments we've noticed up to a factor 3 in training time when switching from the standard 32bit training to the 32bit bitwise training procedure.

---

> > ### Comment · Reviewer_gsV7 · 2021-11-24
> > **Thank you.**
> >
> > Many of my concerns were addressed and I raise my score to 6.

---

### Official Review · Reviewer_MjPw · 2021-11-01

**Correctness:** 1
**Technical Novelty And Significance:** 2
**Empirical Novelty And Significance:** 1
**Recommendation:** 3
**Confidence:** 4

**Main Review:**

Unfortunately, there are serious issues with this submission:

The introduction and motivation seem to be written for a paper that is on pruning not quantization, and they are out of context compared to the title, abstract, and rest of the paper. This is most likely a LaTeX error. The motivation of the paper and comparison to prior art being excluded, it is very hard to assess the quality of the work. I did however try to extrapolate what the authors intended to write, and below is my review for Section 3 and onwards.

In Section 3, what is the expression for \alpha in terms of layer dimensions and number of bits such that the He initialization standard deviation condition is satisfied? This seems like an interesting result that can be added inline in the paper, rather than just implicitly mentioning that an expression for \alpha was derived and used.

What is a He distribution? In (He, 2015), variance engineering is done and the contributions are conditions on the variance of the initialized weights. The distributions themselves are either uniform or normal. Can the author clarify what they mean by He distribution, I believe they mean He conditions on variance.

The experiments are performed on very trivial networks deployed on the MNIST and CIFAR-10 dataset. Can the authors evaluate their work on more contemporary networks such as ResNet on ImageNet and similar tasks?

Message encoding in the neural network's weight using steganography in training is interesting. However, why does it matter? And can these results be generalized on larger networks.

**Summary Of The Paper:**

The paper proposes arithmetic decomposition and training on individual bits in order to achieve low-precision quantization.

**Summary Of The Review:**

Unfortunately, it seems this paper was rushed into submission. The idea sounds interesting, however, the proposed manuscript has several serious issues as raised in my main review. I therefore recommend rejection of this paper.

---

> ### Author Response · Authors · 2021-11-19
> **Response to Reviewer MjPw**
>
> "The introduction and motivation seem to be written for a paper that is on pruning not quantization, and they are out of context compared to the title, abstract, and rest of the paper. This is most likely a LaTeX error. The motivation of the paper and comparison to prior art being excluded, it is very hard to assess the quality of the work. I did however try to extrapolate what the authors intended to write, and below is my review for Section 3 and onwards."
>
> > We are sorry you interpreted our manuscript as a LaTeX compilation error. We assure you it is not. We have extended the motivation section to touch upon quantization and also added a section on the connection with previous works.
>
> "In Section 3, what is the expression for $\alpha$ in terms of layer dimension".
> > We have determined $\alpha$ algorithmically via a simple binary search such that the He variance condition is fulfilled for each layer. We have explicitly added that in the appropriate paragraph.
>
> "Can the author clarify what they mean by He distribution, I believe they mean He conditions on variance."
> > We have clarified this in the main body of the text. Indeed, we mean the He condition on the variance, as described a few sentences earlier in the paragraph.
>
> "Can the authors evaluate their work on more contemporary networks such as ResNet on ImageNet and similar tasks?"
> > We did use ResNet, as explicitly mentioned and referenced in the paper.
> The limited amount of time (and resources) we have for the revision of our work unfortunately does not allow to reach any sensible results on datasets such as ImageNet.
>
> Message encoding in the neural network's weight using steganography in training is interesting. However, why does it matter? And can these results be generalized on larger networks."
> > This training technique directly demonstrates the robustness of neural networks when facing strong biases in the weights, either random noise or structured data.
> We have no reason to believe this technique would completely break down for larger networks or larger datasets.

---

> > ### Comment · Reviewer_MjPw · 2021-11-25
> > **Thanks for the response**
> >
> > Thank you authors. I am afraid the minor clarifications provided do not change my mind that this paper is not ready for publication.

---

### Official Review · Reviewer_LhyK · 2021-11-09

**Correctness:** 2
**Technical Novelty And Significance:** 1
**Empirical Novelty And Significance:** 2
**Recommendation:** 3
**Confidence:** 5

**Main Review:**

The authors provide a nice background of pruning literature and sparse representations in neural networks.  I believe they were trying to provide a background for the idea that sparseness is something inherent and should/could be uncovered by gradient descent.  However, the background citations are for a slightly different, but related topic; pruning techniques are not exactly what this paper is about, but it does share some background with spares representations.

In motivation, for line" One possible explanation is that, at least for classification tasks, the usual cross–entropy loss without ad- ditional regularization techniques are not well suited for this" there needs to be citation.

The description of STE in section 3 needs more description.

The experimental results look very preliminary.  Using only LeNet and CIFAR10 might be a good way to triage a technique, but it is very different to draw any conclusions based on these.  They are too small and the results between them barely suggest any trend.  The authors essentially show a known result that training only sign bits (Ivan and Florian 2020) yields good results.  What additional science have the authors uncovered?

The last section of encoding messages in weights seems quite unrelated to the rest of the paper.  It is interesting, but the results seem random and they don't give us any new insight into the learning process of neural networks.

**Summary Of The Paper:**

The authors propose a method to train individual bits of weights of a neural network.  The central motivation is the question of why doesn't gradient descent "discover" inherent sparsity by setting certain weights to zero? The authors suggest that since there are so many possible states of a 32b integer number, the probability of landing on all zeros is vanishingly small. Using the bit-wise training technique, they are able to demonstrate that good performance can be achieved with few bits of weight representation. The authors also do a series of experiments where only certain bit positions are changeable to demonstrate which positions are most relevant to getting good classification performance.

**Summary Of The Review:**

This paper has some interesting concepts but falls short on uncovering novel insight.  They are able to recapitulate other papers' results, but even this seems a bit tenuous. The experimental support for any conclusions is too weak in this paper to draw any real conclusions.

---

> ### Author Response · Authors · 2021-11-19
> **Response to Reviewer  LhyK**
>
> In motivation, for line" One possible explanation is that, at least for classification tasks, the usual cross–entropy loss without additional regularization techniques are not well suited for this" there needs to be citation.
> > Unfortunately there cannot be a citation since this is the presumption of the authors and one of the issues motivating the development of this training algorithm.
>
> The description of STE in section 3 needs more description.
> > We have extended the discussion on STEs in the corresponding section.
>
> "The experimental results look very preliminary. Using only LeNet and CIFAR10 might be a good way to triage a technique, but it is very different to draw any conclusions based on these. They are too small and the results between them barely suggest any trend. The authors essentially show a known result that training only sign bits (Ivan and Florian 2020) yields good results. What additional science have the authors uncovered?"
>
> > Although not explicitly described in the paper, the bit-wise training technique connects the topics of network pruning and quantization in a single framework. Results shown in works such as BinaryConnect, XNOR-Net, Ternary networks and algorithms such as sign-flipping and edge-popup are just particular instances of our training technique. To the best of our knowledge this is the first work that shows how pruning naturally emerges in neural networks without explicitly masking neuron connections. It also naturally shows that 32bits are superfluous and represent an overparameterization in terms of bit-width.
>
> "The last section of encoding messages in weights seems quite unrelated to the rest of the paper. It is interesting, but the results seem random and they don't give us any new insight into the learning process of neural networks."
>
> > The last section is more a corollary of our training algorithm, and directly shows how overparameterized networks are able to generalize: by leveraging the regularization introduced by the least significant bits of the weights.

---

### Decision · Program_Chairs · 2022-01-20

**Decision:**

Reject

**Comment:**

This paper presents some interesting new ideas on training binary neural networks. However, as many reviewers point out the study is quite limited by their experimental section, and some technical issues were raised. These are criticisms that remain largely unaddressed after the author response, hence the paper is not ready for publication at its current form.